# Effectiveness of Group Parent-Child Interaction Therapy on Problem Behaviors in Chinese Kindergartners

**DOI:** 10.3390/ijerph20043446

**Published:** 2023-02-15

**Authors:** Juanjuan Sun, Mowei Liu, Xiaoyun Li, Yuena Zhou, Yan Li

**Affiliations:** 1Shanghai Early Childhood Education College, Shanghai Normal University, 100 Guilin Road, Xuhui District, Shanghai 200234, China; 2Department of Psychology, Trent University, Peterborough, ON K9J 7B8, Canada

**Keywords:** parenting intervention, externalizing problems, internalizing problems, kindergartner

## Abstract

Problem behaviors in early childhood may put young children at risk for negative behavioral and psychosocial problems. This study examined the effectiveness of group PCIT on Chinese young children’s externalizing and internalizing problems. The participants were 58 mothers with their children aged 2–3 years (M = 2.95 years, SD = 0.22), assigned to an immediate treatment (*n* = 26) group or to a waitlist (*n* = 32) control group. The program involved comprehensive group intervention and featured weekly 60–90-min sessions, totaling ten sessions over three months. Results indicate that group PCIT not only significantly improved teacher-reported problem behaviors in children, but also improved observed maternal parenting behavior. These findings support the use of group PCIT in Chinese children and provide mothers with an evidence-based tool to address problem behaviors in a non-clinical population.

## 1. Introduction

The prevalence of problem behaviors among young children in the general population has attracted significant attention in recent years [1,2]. According to a report released by the Chinese Academy of Sciences, 17.5% of Chinese children aged 6–16 years suffer from mental disorders, a high prevalence rate compared to those reported from other nations across the globe [3,4]. A growing body of evidence suggests that persistent and untreated early problem behavior can result in academic problems [5], conduct disorders in middle childhood and adolescence [6], and psychopathology and anti-social behavior in adulthood [7,8].

Inadequate parenting is an important risk factor that has been associated with a variety of problem behaviors [9,10,11]. Interventions that focus on parenting training have been developed and administered to clinical samples including children with ADHD, ODD, conduct disorders, anxiety, and depression [12]. Moreover, internalizing problems and externalizing problems often co-occur in childhood and adolescence [13,14,15,16]. Because these problems share major risk factors, they may be effectively treated with similar treatment regimens [17]. Thus, intervention for externalizing problems may have secondary benefits for internalizing problems [18,19,20].The present study is dedicated to evaluating the effectiveness of Parent–Child Interaction Therapy (PCIT) on reducing both internalizing and externalizing problems.

### 1.1. Problem Behaviors

Owing much to Achenbach’s work [21], individual functioning and dysfunction is classified along two empirically derived dimensions, namely, internalizing dimension and externalizing dimension. Internalizing dimension includes overcontrolled behaviors such as withdrawal, anxiety, fearfulness, and depression, whereas externalizing dimension includes under-controlled behaviors such as hyperactivity, defiance, and aggression [2,13,22]. In comparison with the categorial approach that emphasizes discrete individual differences, the dimensional approach of psychopathology emphasizes quantitative differences. According to this approach, every child may display problem behaviors to some extent, and mild problems may become more serious problems or clinically diagnosed disorders.

Internalizing and externalizing problems may develop as early as early childhood and are associated with peer problems, parent-child conflict, and other maladjustment [23,24]. Early emotional and behavioral problems may indicate the onset of adverse psychiatric and psychosocial development across the life span [19]. The societal costs of leaving problem behaviors untreated can be enormous [25]. For example, it was found that an estimated $2.3 million could be saved by successful treatment of high-risk youth with disruptive behaviors [26]. Heckman [27] compared the effects of different interventions from early childhood to adolescence and found that the effectiveness and cost-effectiveness of interventions decreased significantly with age. Therefore, in the hope of minimizing future treatment needs and promoting healthy development, it is of great importance to conduct early interventions to reduce risks and the incidence of problem behaviors in a young non-clinical population. 

Although internalizing and externalizing problems have distinctive main characteristics, they are closely related and are likely to co-occur in childhood and adolescence [16,28]. Children with high levels of externalizing problems are more likely to have both concurrent and consecutive high levels of internalizing problems [28]. According to Nivard et al. [16], children’s internalizing and externalizing problem trajectories tend to coincide over time, and nearly half of children exhibit co-occurring problems. Moreover, children who consistently display internalizing and externalizing behavior problems are more likely to suffer from later negative consequences, such as peer rejection, involvement in risky activities, and substance abuse during adolescence [14,29]. When assessing the efficacy of early therapies, however, few studies have taken into account the co-occurrence of behavioral problems. The current study aims to address this limitation in the literature.

### 1.2. Relationships between Parenting and Problem Behaviors

There is a growing body of work showing that the development of problem behavior has been associated with environmental influences, in particular inadequate parenting [10,30,31]. Parenting that involves promoting children’s social development through parental support, guidance and positive control has been mainly related to positive child outcomes such as higher self-worth, and social competence [32,33,34], whereas parenting that is characterized by negative control which limits children’s autonomy has been associated with higher levels of externalizing symptoms and anxiety over time [11,31,35].

The family-centered context in Asian societies further points to the crucial role of parenting in mental health outcomes for Asian children [36]. Traditional Confucian ideologies emphasize family allocentrism, close relationships, and maintaining social order and interpersonal harmony [37,38]. Reflecting these values, Chinese society tends to form an interactive pattern of parental authority and child obedience [39,40]. Compared with North American parents, Chinese mothers scored higher on physical coercion [41]. When their children are disobedient and engaging in disruptive behaviors in public, Chinese mothers tend to use forceful control to demand immediate compliance from the child [42,43]. Nevertheless, Chinese children presumably interpret parental control as a sign of parental involvement and caring concern [32]. Though it is possible that parental control is exercised and interpreted differently across cultures [44,45], a growing body of evidence shows that, within one culture, parental negative control, in comparison with positive control, undermines children’s psychological adjustment [31,35,46]. However, as far as we know, there is no research to explore the effectiveness of PCIT on parental parenting behavior and children’s problems behavior in the context of Chinese culture. Therefore, for the healthy psychological development of children, it is necessary to improve the quality of parent-child interactions, especially for Chinese parents.

### 1.3. Advances in Early Intervention Studies

While cognitive-behavioral therapy (CBT) is effective in treating youth problem behaviors, there are some obstacles for young children [26,37,47]. Because younger children’s cognitive development is still immature, traditional CBT techniques (such cognitive restructuring) may not be suitable for use with them [19,47]. In particular, children may have difficulty understanding, reflecting on, and expressing their inner feelings or thoughts, which could make it difficult to use CBT on them. In contrast, parenting treatments that attempt to improve parent-child interactions and children’s developmental outcomes can be administered from infancy through late adolescence [48]. The effectiveness of parenting interventions for reducing children’s problem behaviors has been widely evidenced [12,49].

Parent-Child Interaction Therapy (PCIT) [50] is an empirically supported and developmentally informed parent intervention focusing on modifying the quality of parent-child interactions and child behavior. It is mainly used for children aged 2–7 years [51]. By creating a positive parental example for children, PCIT employs strategies such as praise and enthusiasm and teaches more effective direct command sequences in an effort to decrease children’s problem behaviors. PCIT consists of two treatment phases. The Child Directed Interaction (CDI) phase aims to enhance the positive parent-child relationship by improving parents’ parenting behavior [52]. At the CDI stage, parents learn to reduce negative parenting behaviors such as questioning, commanding, and criticism while increasing positive parenting behaviors such as “PRIDE” skills, e.g., pride, reflecting, describing and enthusiasm. The Parent Directed Interaction (PDI) phase aims to give effective instructions and maintaining the consistency of instructions [53]. At the PDI stage, parents are taught a corresponding “compliance sequence” and a time-out technique. The standard PCIT involves the therapist provides direct live coaching through a Bluetooth in-ear microphone while they engage with their children.

Parent-training programs mainly includes two formats: individual and group. Previous studies have shown the effectiveness of individually administered PCIT for child problem behavior [50,54,55] Meta-analyses indicate that there is minimal overall difference between the effectiveness of individual and group treatment modalities [56,57,58]. Additionally, minority families prefer group therapy over traditional therapy services [59,60]. Hare and Graziano [61] found that PCIT the large groups is most cost-effective in improving parental parenting behavior and children’s compliance by comparing three different forms of PCIT (intensive PCIT, small group PCIT and large group PCIT). In addition, group therapy may better deal with treatment obstacles, such as high parenting stress, low social support, and limited financial resources, by providing a low potential cost treatment scheme with a larger social support network [59,60]. Therefore, in this study, we implemented group PCIT for the parents of Chinese kindergarteners to make full use of the potential advantages of group PCIT (such as social support, acceptance, and cost-effectiveness).

With the continuous optimization and adjustment of PCIT technology, PCIT has gradually expanded from the initial focus on children with disruptive behavior disorder to a wider group of children [54,55]. For example, researchers began to explore the effectiveness of PCIT on children’s behavior inhibition [62], separation anxiety disorder [63], and a variety of anxiety disorders [64]. Meta-analyses suggest that PCIT is effective for a variety of externalizing problems such as disruptive behavior disorders [65] and internalizing symptoms such as anxiety [19]. Existing studies mainly examine externalizing problems or internalizing problems separately, while few studies explored the effects of PCIT intervention on both internalizing and externalizing problems.

#### The Present Study

In summary, while PCIT is empirically supported as an effective intervention for children with various externalizing and internalizing problems and their parents, existing studies primarily focus on specific types of problem behaviors in clinical samples. Few studies have investigated whether PCIT can reduce both externalizing and internalizing problems at the same time in a young non-clinical population. The present study represents a prevention effort that focuses on reducing negative parenting and externalizing and internalizing problems in normal children. Specifically, the purposes of this study were to examine (1) whether PCIT would significantly reduce children’s externalizing and internalizing problems at the same time; and (2) whether PCIT intervention can improve the quality of parent-child interaction. Based on the literature reviewed, we hypothesized that (1) children in the PCIT group would display fewer problem behaviors, including internalizing and externalizing problems; and (2) parents who received group PCIT intervention would significantly improve the quality of parent-child interaction as demonstrated by increased positive parenting behavior and decreased negative parenting behavior. Moreover, parents who took part in the PCIT intervention demonstrated high satisfaction with the intervention’s results.

## 2. Materials and Methods

### 2.1. Participants

A statistically significant medium effect (2 × 2 design) with *η*^2^ = 0.13, and α = 0.05 requires approximately 46 participants to attain 95% power. The current study originally recruited 67 children and their mothers from kindergarten (see Figure 1). Two children were excluded due to medical conditions. Three mothers declined to participate in the intervention. Four mothers allocated to intervention did not participate/complete the intervention. Data from 58 children (M = 2.95, SD = 0.22) and their mothers were included in the final analyses. The sample (58) provided sufficient power to detect a medium effect size (*η*^2^ = 0.13). 

### 2.2. Procedure

The research ethics board examined and approved the current study. Research information and recruitment letter were posted on the participating kindergarten website. Informed consent was obtained from mothers for them and their children’s participation in the study. Mothers completed a family demographic questionnaire including details about their child’s age, gender, number of siblings, maternal age, maternal level of education, and family structure. Mother-child dyads were invited to the kindergarten lab for a pretest of parenting behaviors (positive control and negative control; see Section 3) observed from mother-child interactions in a 15 min origami session. We videotaped mother-child interactions using a single camera. Teachers were asked to complete the child behavior checklist for 2 to 3-year-old children. Participants were then assigned to an immediate treatment group (IT) or a waitlist control group (WL). The IT group received a ten-week-PCIT intervention. Immediately after the IT group completed the intervention, parenting behaviors and child problem behaviors were assessed again using measures identical to those used in the pretest for both groups. In addition, mothers from the IT group were asked to complete the therapy attitude inventory to evaluate their experiences and satisfaction with the intervention received. 

### 2.3. Intervention

Chinese parents typically adopt negative control strategies that are more structured, punitive, and hostile, with great emphasis on parental monitoring of children’s behavior [66,67,68]. PCIT emphasizes avoiding negative control and providing positive control and guidance to children. Mothers in the IT group received ten weekly PCIT sessions delivered by a trained teacher. Each session was 60–90 min long. In addition, mothers should practice their PCIT skills at home in a 5 min “special time”. All sessions were completed by a team composed of doctoral, senior psychological counselors and graduate assistants.

Mothers were instructed CDI and PDI skills during the training sessions. The first three lessons focus on specific “do” skills (i.e., praise, reflection, imitation, description, and enthusiasm). The next two lessons focus on “don’t” skills (i.e., commands, questions, and criticism) to avoid overcontrolling children. The last five sessions focused on “PDI” skills including providing effective command via lecture, modeling, and role-play. Each session started with parenting skill instruction followed by demonstration and practices. After the teacher demonstrated certain PCIT skills with a mother volunteer, mothers were provided with toys and encouraged to practice PCIT skills with a partner for 10–15 min (with each mother role-playing either the parent or the child). The coach would monitor and provide positive reinforcement and guidance throughout the practices. Handouts summarizing PCIT skills were provided at the end of each session.

## 3. Measures

### 3.1. Demographic Questionnaire

Caregivers completed a demographics questionnaire at the pre-treatment assessment to gather information related to caregiver and child, and their household: children’s age, gender, and only-child status (only child or non-only child); mother’s age and educational level (i.e., high school graduate or below, college education, bachelor’s degree, and master’s degree or above), and family structure (i.e., nuclear family, stem family, joint family, and others).

### 3.2. Problem Behavior

The child behavior checklist for 2 to 3-year-old children (CBCL/2–3) [69] was a 100-item completed by teachers during the pretest and posttest. The teachers were blind to the goal of the study and the classification of children. Teachers were asked to report on the frequency of problem behaviors in children on a 3-point scale (0 = never to 2 = often). The scores from the two broadband scales (internalizing and externalizing behaviors) were summed as a measure of children’s symptom severity. In this study, we also assessed four narrowband problem behaviors: aggression (31 items), destruction (14 items), social withdrawal (14 items), and depression (14 things) to determine whether the intervention changed specific features of problem behavior. Internal reliabilities for internalizing and externalizing problem behaviors subscales were acceptable in this study (*α*_pretest_ = 0.80 and 0.69, respectively; *α*_posttest_ = 0.86 and 0.82, respectively).

### 3.3. Parenting Behaviors

Parent-child interaction were observed in the laboratory sessions during an origami paper-folding task [70,71]. Parents were asked to teach their child to complete an origami paper-folding task which they enjoy within 15 min. Four independent observers who blinded to the study hypothesis double coded fifteen percent of the interaction videos. The average score for each scale was calculated by dividing the total number of one-minute time sample units by the sum of the ratings for each scale. Cohen’s Kappas ranged from 0.81 to 0.93.

#### 3.3.1. Maternal Positive Control

Maternal positive control was defined as well-timed, active, and positive control behaviors that facilitate the child’s competent functioning. Maternal positive control was evaluated on a 3-point scale, where a score of 1 meant “none”; a score of 2 indicated “moderate positive control” (i.e., mother determines or chooses the activity for the child but then allows child time to adjust to the activity: ‘the parent asks the child to choose the color of origami paper or origami model’); and a score of 3 indicated “outright positive control” (i.e., scaffolding by verbally assisting the child, explaining the activity, elaborating and expanding on the task, and provides guidance: ‘the dotted lines here tell us where we have to fold the paper so that it looks like…’).

#### 3.3.2. Maternal Negative Control

Maternal negative control refers to parents interfering excessively with their children’s behavior and thoughts against their wishes. Each minute of the mother-child interaction was rated on a 3-point scale, with a score of 1 indicating “none”; a score of 2 indicating “moderate negative control” (i.e., verbally intrusive child: ‘excessive instruction or directions’); and a score of 3 indicating “outright negative control” (i.e., frequent unnecessary dictatorial instructions/physical intrusiveness: ‘grabbing the paper from the child to demonstrate use of paper’).

### 3.4. Maternal Intervention Experiences and Satisfaction

After completing the intervention, mothers in the immediate treatment group were asked to complete the therapy attitude inventory (TAI) [72]. The TAI is a 5-point scale that measures parental satisfaction with the treatment procedure and outcomes (1 = severely dissatisfied, 5 = extremely satisfied). The higher the score indicate the higher level of maternal satisfaction with the intervention. The Cronbach’s alpha was α = 0.93 in the present study. 

## 4. Data Analytical Strategy

SPSS 26.0 was used for all statistical analysis. Mean substitution was used to deal with missing data. The first step was to conduct a one-way analysis of variance (ANOVA) to compare the demographics and pre-treatment problem behavior between the intervention and waiting groups. Children’s only child status, family structure and teachers’ teaching age are significantly related with children’s problem behavior, so they are used as control variables. To assess the effectiveness of the invention (child problem behaviors and maternal parenting behaviors), a 2 × 2 repeated measures ANOVA was carried out with time (pre- and post-treatment) as a within-subjects factors and experimental condition (WL, IT) as a between-subjects factor. Furthermore, partial *η*^2^ calculations for effect size estimations for overall ANOVAs were made to compare the impact of interventions. Effect sizes interpretation follows recommendations by [73]; small (d = 0.10, *η*^2^ = 0.02), medium (d = 0.50, *η*^2^ = 0.13), and large (d = 0.80, *η*^2^ = 0.26).

## 5. Results

### 5.1. Preliminary Analyses

The demographic characteristics and pre-treatment baseline on outcome measures of all families initially allocated to WL or IT are compiled in Table 1. There was no significant difference between the waiting group and the intervention group in terms of demographic factors, children’s problem behavior, and mothers’ parenting behavior (see Table 1 and Table 2). 

### 5.2. Intervention Effects of Problem Behavior

A repeated measures ANOVA with experimental condition as between-subjects factor and time as within-subjects factor was performed to determine if any changes of overall problem behaviors could be connected to the treatment. Only child status, family structure and teaching age were taken as covariate variables. The interaction between experimental condition and time was statistically significant, *F* (1, 49) = 10.82, partial *η*^2^ = 0.181 (see Table 2). For children in the IT group, the intervention was effective in reducing overall problem behaviors, *F* (1, 49) = 20.43, partial *η*^2^ = 0.294, whereas for children in the WL group, problem behaviors were not significantly changed, *F* (1, 49) = 0.00, partial *η*^2^ = 0.000. A significant difference in the intervention group was evident in a priori comparisons of the mean change scores from the pretest to the posttest, *t* = 5.02, *p* < 0.001.

When exploring whether the intervention had an impact on specific problem behavior (social withdrawal, depression, aggressive or destructive behaviors), we found a significant two-way interaction effect between experimental condition and time, *F* (4, 46) = 2.96, partial *η*^2^ = 0.205. The intervention was effective in decreasing child aggressive behavior, *F* (1, 49) = 11.77, partial *η*^2^ = 0.194, but not for destructive behavior (*p* = 0.130), depressive behavior (*p* = 0.074), and social withdrawal (*p* = 0.144).

### 5.3. Intervention Effects of Parenting Behaviors

We performed a repeated measures MANOVA with experimental condition as a between-subjects factor and time as a within-subject factor to assess intervention effects on maternal parenting behaviors. There was an interaction effect between experimental condition and time, *F* (2, 53) = 3.56, partial *η*^2^ = 0.118 (see Table 3). For maternal negative control, there was a significant interaction between experimental condition and time, *F* (1, 54) = 7.24, partial *η*^2^ = 0.118. Compared with the waitlist group, maternal negative control in the intervention group decreased significantly after the intervention, *F* (2, 53) = 3.38, partial *η*^2^ = 0.113. A priori contrasts of the mean change scores in maternal negative control from pretest to posttest showed a significant difference in intervention group, *t* = 0.07, *p* < 0.05.

However, for maternal positive control, there was not a significant interaction, *F* (1, 54) = 0.51, partial *η*^2^ = 0.009. In addition, mothers reported high satisfaction with the intervention process and child problem behavior change with an average score of 42.29 (range 32–50).

## 6. Discussion

The goal of this study was to investigate the effectiveness of group PCIT program on improving Chinese young children’s problem behavior and maternal parenting behaviors, as measured by teacher-reported CBCL, and parent-child interaction observation, respectively. The results indicated that mothers who received PCIT intervention displayed lower level of negative control behaviors in mother-child interaction. The overall problem behaviors and aggression in children from the intervention group were significantly reduced. 

According to the results of the posttest, the IT group’s overall problem behaviors in children were significantly lower than those of the WL group, supporting Hypothesis 1. This suggests that the group PCIT was successful in reducing children’s problem behaviors in general and aggression. This may be due to the co-occur internalizing and externalizing problems and the presence of shared risk factors, such as conflict parent-child relationship, negative parenting behavior, and high parental stress [47]. The comorbidity between externalizing problems and internalizing problems have been consistently documented in both clinical and non-clinical samples [1,2,3]. It has been argued that the comorbidity might be due to mutual cause and effect or a common underlying causal factor [74]. Previous intervention studies focused on either internalizing or externalizing problems separately, however, it is reasonable to examine whether an intervention program is effective for both internalizing and externalizing problems at the same time given their common co-occurrence. Moreover, based on the dimensional model of psychopathology, internalizing behaviors and externalizing behaviors are continuous dimensions. For both dimensions, a child would be diagnosed with a clinical disorder when he/she presented extremely high levels of symptoms or a pattern of symptoms that indicate severe impairment. Children who are considered as “normally” behaved may show problem behaviors to an extent that does not meet the diagnostic criteria. It is possible the mild symptoms may become severe and problematic if left untreated. As a result, the main goal of the current study was to determine the effectiveness of group PCIT on reducing both internalizing and externalizing problems in a non-clinical Chinese population. In addition, the change in children’s problem behavior can be due to the parent training provided to mothers in the intervention group. As PCIT directly modifies parenting behavior and the pattern of parent-child interaction, the improvement of parent-child interaction may result in a decrease in problem behaviors in children.

Consistent with Hypothesis 2, compared to the WL group, maternal negative control of the IT group was significantly decreased at the posttest, indicating that the group PCIT effectively reduced mother’s negative parenting behavior. Since the primary objective of PCIT is to improve parent-child relationships, namely by lowering negative parenting behaviors, this effect size on parenting skill may be the most significant [61]. Similar to the Chinese saying, “Beating and scolding is the emblem of love”, traditional Chinese parents see negative control as a display of parental participation and concern [42,75]. Wang and Liu found that harsh [75] discipline among parents of children ages 3 to 15 is quite common in modern China. Especially, mothers as primary caregivers in their families are more likely to apply harsh discipline than fathers. According to the social learning perspective, negative parenting behaviors leads to a coercive cycle between parents and children, and these negative interactions will lead children to learn to use negative control strategies not only in the family, but also in their peers. When parents use negative controls, children can learn harsh, coercive, and physical dominance as a primary way to manage social interaction [76,77]. In contrast, when parents use positive controls, children can learn caring and supportive behaviors to manage social interaction. 

Bowlby believe that infant-mother attachment relationships set the stage for psychological development later in life [78]. Based on the interactive experiences with their primary caregivers, infants develop a set of mental representations of themselves, others, and the character of human relationships (internal working models). Mothers who are sensitive, responsive, and supportive are more likely to have securely attached infants who construct positive internal working models of themselves and others, whereas mothers who are insensitive, unresponsive, cold, and hostile are more likely to have infants who form negative views of themselves and/or others [79]. Maternal sensitivity, responsiveness, and affection are important defining features that differentiate positive control and negative control in the present study. Mothers who use positive control are responsive to their child’s needs. Expectations for their children are clearly explained and elaborated. Assistance and guidance are provided when needed. In contrast, mothers who use negative control are generally insensitive or less responsive to their child’s needs. Parental dominance and child conformity are emphasized through power-assertive, prohibitive, and punitive strategies with their children. Myriad meta-analyses have identified negative control as a robust predictor of the most common mental health problems in children and adolescents, including internalizing problems such as depression, anxiety, and externalizing problems such as aggressive behavior [80].

To reduce the incidence of problem behaviors, PCIT seeks to enhance parent-child attachment and to help parents learn effective strategies to provide commands. Consistent with our hypothesis, there was a significant decrease in maternal use of negative control after PCIT in the IT group. Specifically, during the mother-child interactions, mothers decreased their overcontrolling behaviors, such as excessive commands and questions. This finding is consistent with previous studies, confirming a positive effect of parenting training on decreasing maternal negative control [59,81]. 

This study did not find significant improvements in maternal positive control in immediate invention group compared with the waitlist group, indicating no significant changes after the parenting training, which failed to provide supplementary evidence to the existing literature [82]. It is possible that positive control and guidance may require more parenting knowledge and time to use the skills learned. Future studies may extend the intervention to a longer period to allow parents to have more time to acquire and apply positive parenting skills. 

To summarize, the findings of this study contribute to the literature by proving that PCIT intervention is effective to non-clinical Chinese kindergartners. Overall, problem behaviors, aggression, and depression in children were reduced after the intervention. Mothers from the intervention group displayed lower level of negative controls. The results suggest that PCIT can be effective when administered in a group setting. 

### Limitations and Strengths

Although these are encouraging results, there are still limitations in this study. First, although the effectiveness of PCIT in China was demonstrated, no further follow-up assessments were conducted. Evidence from previous studies shows that the benefits obtained through personal PCIT can be sustained [83]. Long-term positive outcomes can be observed at a 6-week follow-up [81] and may persist up to 3 years [84]. Future studies should examine the long-term treatment effects by conducting follow-up assessments in different contexts. 

The other limitation is about the quasi-experimental design. No random assignment of the research participants was used. Although the intervention group and the control group were similar at the beginning in many ways, causal connections between the intervention and behavioral changes in mothers and children need to be further established. Interpreting the findings should be performed with caution. 

Despite the limitations in our methodology, the present study investigates further the efficacy and portability of the application of PCIT in a group setting. In this study, we analyzed treatment outcomes from multiple methodologies, including teachers’ reports on children’s problem behavior, direct parent-child interaction observation, and self-reported levels of satisfaction. Our pilot study appears to have resulted in beneficial improvements among the families who participated. Compared with the data provided by parents’ self-report, the observation of parent-child interaction can truly observe the changes in parental rearing behavior after PCIT intervention [85]. 

The effectiveness of PCIT in addressing both externalizing and internalizing problems is a significant strength of the current study. The previous meta-analysis on PCIT has independently examined the impact of externalizing behavior [65] and internalizing behavior [19]. This study expands earlier studies, validates the use of PCIT to address externalizing and internalizing problems. 

Given that group PCIT is an effectiveness treatment for non-clinical children’s internalizing and externalizing problems, it can bring more treatment options for children. Moreover, it may be a financially viable service delivery model for school intervention. As a parent-led evidence-based intervention, PCIT can be applied to preventing and treating younger children’s problem behavior. PCIT can play up the role of parents to encourage and support children’s positive actions, which may be able to solve some of the problems in current golden standard therapy (CBT) [19,47]. Considering the increased resources needed to treat numerous families at once, it is questionable whether the group format is cost-effective [86], but its potential to improve attendance, treatment completion rate, and treatment satisfaction is worth examining.

In addition, our data could provide reference for PCIT in the future prevention and intervention of schools. More and more countries and scholars are promoting and implementing PCIT services [59], but previous studies have mainly focused on clinical children. This study has proved the effectiveness of PCIT on non-clinical children’s problem behaviors and provided researchers with data on the effectiveness of PCIT in “real world customers”.

## 7. Conclusions

In sum, group PCIT intervention program has proven to be effective in decreasing maternal negative control behaviors and child problem behaviors in China. Therefore, PCIT should become a crucial component of early childhood intervention for problem behaviors.

## Figures and Tables

**Figure 1 ijerph-20-03446-f001:**
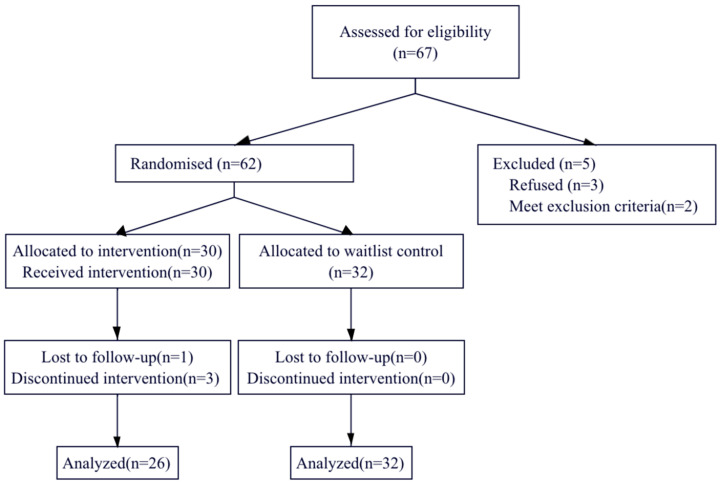
Participant flowchart.

**Table 1 ijerph-20-03446-t001:** Demographic information for the total sample.

Means (SD) or Percentages				
	Total (*n* = 58)	WL (*n* = 32)	IT (*n* = 26)	*p*
Child age	2.95 (0.22)	3 (0.01)	2.88 (0.33)	0.083
Child gender—male%	51.70%	46.90%	57.70%	0.421
Only-child%	63.80%	53.10%	76.90%	0.062
Mothers’ age(years)	35.72 (4.46)	36.43 (4.69)	34.83 (4.35)	0.193
Mothers’ education-Bachelor’s degree%	90.70%	93.30%	87.50%	0.586
Family structure-nuclear%	41.40%	37.50%	46.20%	0.314
Teachers teaching years	10.33(8.10)	10.31(8.17)	10.35(8.18)	0.988

**Table 2 ijerph-20-03446-t002:** Means for children’s problem behavior and between-group difference in pre-post.

	Waitlist Group (*n* = 32)		Immediate Treatment Group (*n* = 26)		
Measures	Pre	Post	Pre	Post	*p*
Social withdraw	14.41 ± 1.78	13.94 ± 1.50	15.23 ± 2.83	14.59 ± 1.79	0.144
Depression	14.38 ± 0.71	14.28 ± 0.63	14.62 ± 1.06	14.24 ± 0.71	0.074
Aggressive	32.93 ± 2.56	32.47 ± 2.68	34.15 ± 3.59	31.93 ± 1.16	0.001 *
Destructive	14.41 ± 0.67	14.25 ± 0.51	14.81 ± 1.41	14.81 ± 1.41	0.130
Overall problem behaviors	75.81 ± 4.22	74.94 ± 4.44	78.19 ± 2.86	75.19 ± 2.86	0.035 *

* *p* < 0.05.

**Table 3 ijerph-20-03446-t003:** Means for parenting behavior and between-group difference in pre-post.

	Waitlist Group (*n* = 32)		Immediate Treatment Group (*n* = 26)		
Measures	Pre	Post	Pre	Post	*p*
Positive control	1.57 ± 0.40	1.82 ± 1.07	1.55 ± 0.30	1.86 ± 0.20	0.480
Negative control	1.03 ± 0.10	1.07 ± 0.14	1.03 ± 0.07	0.97 ± 0.20	0.090

## Data Availability

Data available on request due to restrictions (e.g., privacy or ethical). The data presented in the study are available on request from the corresponding author. The data are not publicly available due to privacy and ethical guidelines.

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
