# Peer review of "Effectiveness of Group Parent-Child Interaction Therapy on Problem Behaviors in Chinese Kindergartners"

_ijerph, 2023, doi:10.3390/ijerph20043446_

Round 1

Reviewer 1 Report

The paper presents an important RCT with PCIT on problem behaviours in 58 Chinese kindergarteners. The researchers used a wait-list design, the intervention was group-based and totalled 10 sessions. Results were impressive, showing that behaviour problems decreased and observed  parenting improved.

This type of work is crucially important for theory development, I.e. establishing causal associations, and to lead to practical applications if replicated. From China not much of such work has been seen in recent years, so this is a welcome addition to our knowledge base.

There are some issues that might be resolved in a revision to enhance the impact of the RCT.

 First, the authors might present a flow chart with detailed reports of inclusion and exclusion of participants in various stages of the RCT.

Second, mean substitution is used where multiple imputation might do a better, less conservative job.

Third, it is not completely clear whether an intent-to-treat has been used, with a robustness test of complete cases.

Fourth, somewhere in one of the tables a mean value of 3 with a SD of 0.00 has been reported which would mean no variation, I.e. no variable to be used.

Fifth, it is unclear why the authors used the covariates they have chosen: “Only child status, family structure and teaching age were taken as covariate variables.” They should also report the robustness analysis without any covariate (this is an RCT anyway).

Sixth, marginally significant results should be considered non-significant according to the NHST tradition.

Seventh, I do not understand how a t = 0.07 can be significant?

Eight, a correlation matrix between all variables might be added as a supplement or in the main text.

Ninth, the starting sentences of the Discussion seem odd, and misplaced, or a wrong copy/paste.

Overall: this is a pilot study which should be replicated in a larger sample and with pre-registration before being applicable in practice.

But this is a great start of a research program on such interventions.

Author Response

Point 1: The authors might present a flow chart with detailed reports of inclusion and exclusion of participants in various stages of the RCT.

Response 1: Thank for your suggestion, we have added a flow chart as suggested. Please see Figure 1 in page 6.

Point 2: Mean substitution is used where multiple imputation might do a better, less conservative job.

Response 2: Thank for your suggestion, we will adopt multiple imputation methods for future data analysis. In this study the missing values were randomly distributed across items and participants, they were substituted with the mean score on the variable for children with the same sex, age, parental educational level, and experimental condition, as a conservative imputation method (Tabachnick & Fidell, 2001), to uniformly include the total set of 58 children in the analyses. Results were similar when missing data were excluded from the analyses and when the imputation strategy of expectation maximization (Tabachnick & Fidell, 2001) was applied.

Point 3: it is not completely clear whether an intent-to-treat has been used, with a robustness test of complete cases.

Response 3: Yes, all participants were fully informed of the study procedures and gave their consent to participate prior to their enrollment. With the strong support of kindergarten and parents, 26 mothers in the intervention group completed all the courses.

Point 4: somewhere in one of the tables a mean value of 3 with a SD of 0.00 has been reported which would mean no variation, I.e. no variable to be used.

Response 4: Thank you for pointing out this problem. We have checked the data and modified the result. Please see Table 1 in page 9.

Point 5: it is unclear why the authors used the covariates they have chosen: “Only child status, family structure and teaching age were taken as covariate variables.” They should also report the robustness analysis without any covariate (this is an RCT anyway).

Response 5: Thank you for this suggestion. We now have added the reason for the control variable.

“In this study, children’s gender and age are not significantly related to children’s problem behavior, while children’s only child status, family structure and teachers’ teaching age are significantly related, so they are used as control variables.” Please see the details on page 9.

Point 6: marginally significant results should be considered non-significant according to the NHST tradition.

Response 6: Thank you for pointing out this problem. We have revised this result as non-significant as suggested.

Point 7: I do not understand how a t = 0.07 can be significant?

Response 7: Thank you for pointing out this problem. We have revised this result as non-significant. Please see page 10.

Point 8: a correlation matrix between all variables might be added as a supplement or in the main text.

Response 8: Thank you for your suggestion. This study focuses on the changes of mother's parenting behavior and children’s problem behavior between the intervention group and the waiting group before and after the intervention. Combining with the prior studies (Hare & Graziano, 2020; Meynen et al., 2022), the correlation coefficient between variables is not the analysis content of this study, so it is not provided.

Point 9: the starting sentences of the Discussion seem odd, and misplaced, or a wrong copy/paste.

Response 9: Thank you for pointing out this problem. We have deleted this sentence.

Reviewer 2 Report

The authors address a relevant topic that requires further research. This reviewer has two main concerns that need to be addressed by the authors for their paper to be acceptable for publication. Further minor issues are also commented in order to improve the paper.

One of the main concerns of this reviewer is related to the advances of the paper. For this purpose, the authors should specify which papers already has studied group PCIT jointly on internalizing and externalizing behaviors. This is not a long-term effects study, neither RCT; therefore the authors must justify why this study is novel. This reviewer is not familiar with Asian literature, maybe this topic has been less researched in this population. The discussion could be improved according to this new information for the introduction section.

The other main concern for this reviewer is related to the statistical power of the analyses, due the small sample size, and the representativeness of the sample. Why this study is relevant in terms of population? Are their results generalizable? The authors must justify the sample size explaining pre or ad hoc power analysis (e.g., G power analysis). The authors must explain better the sampling procedure and justify the relevance of examining that sample. Is a convenience sample? Is representing the population?

If these two main concerns are adequately solved and justified in the paper, minor issues can be addressed and the paper can be considered for publication in opinion of this reviewer. Other minor issues that should be considered are the following:

-The keyword “parent-mediated intervention” could be better formulated as “parenting intervention” to facilitate the searching of the article in the future.

-The introduction includes a summary of information that is latter presented. In opinion of this reviewer, this should be avoided, and that information should be integrated avoiding repetition. The information in the introduction could be better integrated.

-Following current international guidelines for publication of evaluation of interventions, explicit information about the mechanisms of change of the intervention should be included in the introduction section.

-Following those guidelines, more information about the intervention should be provided in 2.3, namely information about the context of the intervention as well as the trainers.

-The first paragraph of the measures section should be deleted. This information is for authors, not for the final version of the paper.

-Some information for the measures is missing in opinion of this reviewer, according to APA guidelines. For example, ad hoc measures should be named as those; number of total items is missing in some questionnaires; type of questions is missing in 3.1.

-Information about missing dealing strategies and statistical assumptions is missing in section 4. In this section there is no reference to MANOVA analyses, although those analyses are later reported. Power analysis should be explained here. There is no information about qualitative analyses performed to codify observations.

-If demographic comparisons is included in 5.1, this should not be stated in the participants section (avoid repetition). Comparison among behavior problems between groups at pretest is desiderable.

-APA style should be reviewed when reporting statistics.

-Following APA guidelines, information should not be repeated in tables and in text. Thus, F and p values are not needed in the text, only eta values. It could be explained if small/medium/large effect size is obtained.

-Why child status, family structure and teaching age were included as covariates? Why not child gender and age, both HIGH relevant variables for problem behavior? What child status means?

-In Table 2, destructive measure is indicated as significant, but p value is .130. Is this correct?

-The first paragraph of the discussion section should be deleted, as these are instructions for authors, not valid for the final version of the paper.

-The improvement of the introduction should end into an improvement of the discussion. If comparison between groups in pretest behavior problems are considered, this could support the discussion (line 433).

This reviewer thanks the authors the opportunity to review this interesting work and encourage to consider these comments that hopefully will be useful for the authors.

Author Response

Point 1: One of the main concerns of this reviewer is related to the advances of the paper. For this purpose, the authors should specify which papers already has studied group PCIT jointly on internalizing and externalizing behaviors. This is not a long-term effects study, neither RCT; therefore, the authors must justify why this study is novel. This reviewer is not familiar with Asian literature, maybe this topic has been less researched in this population. The discussion could be improved according to this new information for the introduction section.

Response 1: First, at present, the PCIT research has been proved to be effective for internalizing and externalizing problem behaviors respectively (Furukawa et al., 2018; Nieter et al., 2013), but the research focuses on one aspect of problem behaviors. Few studies also investigate the effectiveness of intervention on internalizing and externalizing problem behaviors (Please see page 1). Secondly, as far as we know, this is the first experimental report that examines the effects of PCIT on children in China (please see page 2). Finally, previous studies have shown that group PCIT intervention is better due to its unique advantages (Hare &Graziano, 2020; please page 3). Therefore, this study adopted the group PCIT design and explored the effectiveness of intervention on internalizing and externalizing problem behaviors.

Point 2: The other main concern for this reviewer is related to the statistical power of the analyses, due the small sample size, and the representativeness of the sample. Why this study is relevant in terms of population? Are their results generalizable? The authors must justify the sample size explaining pre or ad hoc power analysis (e.g., G power analysis). The authors must explain better the sampling procedure and justify the relevance of examining that sample. Is a convenience sample? Is representing the population?

Response 2: Thank you for pointing out this problem. A statistically significant medium effect (2×2 design) with η2 = 0.13, and α = 0.05 requires approximately 46 participants to attain 95 % power (Faul et al., 2007). The sample (58) provided sufficient power to detect a medium effect size (η2 = 0.13).

Point 3: The keyword “parent-mediated intervention” could be better formulated as “parenting intervention” to facilitate the searching of the article in the future.

Response 3: Thank you for this suggestion. We have revised the keyword “parenting intervention”.

Point 4: The introduction includes a summary of information that is latter presented. In opinion of this reviewer, this should be avoided, and that information should be integrated avoiding repetition. The information in the introduction could be better integrated.

Response 4: Thank you for your suggestion. We have now revised the foreword and deleted the repeated contents. Please see this information on page 2.

Point 5: Following current international guidelines for publication of evaluation of interventions, explicit information about the mechanisms of change of the intervention should be included in the introduction section.

Response 5: Thank you for your suggestion. We have added the information about the mechanisms of change of the intervention in the introduction section. Please see this information on page 4.

Point 6: Following those guidelines, more information about the intervention should be provided in 2.3, namely information about the context of the intervention as well as the trainers.

Response 6: Thank you for your suggestion. We have added more information about the context of the intervention as well as the trainers. Please see this information on page 7.

Point 7: The first paragraph of the measures section should be deleted. This information is for authors, not for the final version of the paper.

Response 7: Thank you. We have deleted the first paragraph of the measures section. Please see page 7. “All sessions are completed by a team composed of doctoral, senior psychological counselors and graduate assistants.”

Point 8: Some information for the measures is missing in opinion of this reviewer, according to APA guidelines. For example, ad hoc measures should be named as those; number of total items is missing in some questionnaires; type of questions is missing in 3.1.

Response 8: Thank you for pointing out this problem. We have more information about measures in 3.1. Please see page 7.

Point 9: Information about missing dealing strategies and statistical assumptions is missing in section 4. In this section there is no reference to MANOVA analyses, although those analyses are later reported. Power analysis should be explained here. There is no information about qualitative analyses performed to codify observations.

Response 9: Thank you for your suggestion. We have elaborated on the implications of the present study in the study section. Please see this information on page 6.

“The mother’s parenting behavior was coded in 1 minute using the time sampling technique. The average score for each scale was calculated by dividing the total number of one-minute time sample units by the sum of the ratings for each scale.” Please see this information on page 8.

Point 10:If demographic comparisons is included in 5.1, this should not be stated in the participants section (avoid repetition). Comparison among behavior problems between groups at pretest is desiderable.

Response 10: Thank you for your suggestion. We have deleted this in the participant’s section. Please see this information on page 5.

Point 11:APA style should be reviewed when reporting statistics.

Response 11: Thank you for your suggestion. We have revised the reporting statistics according to APA styles.

Point 12:Following APA guidelines, information should not be repeated in tables and in text. Thus, F and p values are not needed in the text, only eta values. It could be explained if small/medium/large effect size is obtained.

Response 12: Thank you for your suggestion. We have deleted the repeated information. Please see this information on page 9 and page 10.

Point 13:Why child status, family structure and teaching age were included as covariates? Why not child gender and age, both HIGH relevant variables for problem behavior? What child status means?

Response 13: Thank you for this suggestion. We now have added the reason for the control variable. Please see the details on page 9. Child status means the children is only child or non-only child. Because China has implemented the one-child policy for a long time in the past, there are still some one-child families in China.

Point 14:In Table 2, destructive measure is indicated as significant, but p value is .130. Is this correct?

Response 14: Thank you for pointing out this problem. We have revised this result in Table 2.

Point 15:The first paragraph of the discussion section should be deleted, as these are instructions for authors, not valid for the final version of the paper.

Response 15: Thank you for pointing out this problem. We have deleted the first paragraph of the discussion section.

Point 16:The improvement of the introduction should end into an improvement of the discussion. If comparison between groups in pretest behavior problems are considered, this could support the discussion (line 433).

Response 16: Thank you for your suggestion. We have revised this in the discussion section. Please see this information on page 13.